# Identification of drug-like molecules targeting the ATPase activity of dynamin-like EHD4

**Saif Mohd[1,2], Andreas Oder[3], Edgar Specker[3], Martin Neuenschwander[3], Jens Peter Von Kries[3], Oliver Daumke[1,2]***

**1** Structural Biology, Max Delbrück Center for Molecular Medicine in the Helmholtz Association, Berlin, Germany, **2** Institute of Chemistry and Biochemistry, Freie Universität Berlin, Berlin, Germany, **3** Screening Unit, Leibniz-Forschungsinstitut für Molekulare Pharmakologie, Berlin, Germany

\* oliver.daumke@mdc-berlin.de

**Data Availability Statement:** All relevant data are within the manuscript and its Supporting Information files.

**Funding:** "We thank Deutsche Forschungsgemeinschaft (SFB958, project A12, to

## Abstract

Eps15 (epidermal growth factor receptor pathway substrate 15) homology domain-containing proteins (EHDs) comprise a family of eukaryotic dynamin-related ATPases that participate in various endocytic membrane trafficking pathways. Dysregulation of EHDs function has been implicated in various diseases, including cancer. The lack of small molecule inhibitors which acutely target individual EHD members has hampered progress in dissecting their detailed cellular membrane trafficking pathways and their function during disease. Here, we established a Malachite green-based assay compatible with high throughput screening to monitor the liposome-stimulated ATPase of EHD4. In this way, we identified a drug-like molecule that inhibited EHD4's liposome-stimulated ATPase activity. Structure activity relationship (SAR) studies indicated sites of preferred substitutions for more potent inhibitor synthesis. Moreover, the assay optimization in this work can be applied to other dynamin family members showing a weak and liposome-dependent nucleotide hydrolysis activity.

## Introduction

Dynamin superfamily proteins are involved in various cellular membrane remodeling events [1]. They harness energy from nucleotide hydrolysis to perform mechano-chemical work. The proteins oligomerize on the surface of membranes, leading to stimulation of their low basal nucleotide hydrolysis activity [2]. While most dynamin superfamily proteins are GTP-binding and hydrolyzing enzymes, Eps15-homolgy domain-containing proteins (EHDs) comprise a highly conserved dynamin-related ATPase family, with four members in eukaryotes (EHD1-4) [3]. As other dynamin superfamily members, EHDs bind to liposomes and oligomerize in ring-like structures at their surface, inducing liposome tubulation [4–9]. EHD oligomerization leads to an ~10-fold stimulation of the low intrinsic ATPase activity [4, 7, 9].

EHDs are implicated in a variety of cellular membrane trafficking pathways. For example, EHD1 and EHD3 localize to the endocytic recycling compartment and mediate the release of cargo molecules en route to the cell surface [10–13]. The proteins also play a role in ciliary membrane trafficking and sonic hedgehog signaling [14, 15]. EHD2 localizes to caveolae, which are bulb-shaped invaginations of the plasma membrane [5, 16, 17]. EHD2 is thought to

O.D.) and MDC for their funding and support. "The funders had no role in study design, data collection and analysis, decision to publish, or preparation of the manuscript.

**Competing interests:** The authors have declared that no competing interests exist.

assemble into a ring-like oligomer at the neck of caveolae, thereby stabilizing them to the plasma membrane [18, 19]. EHD4, formerly known as Pincher, mediates retrograde endosomal trafficking of neurotrophin receptors [20, 21] and also participates in primary ciliogenesis [22].

Dysregulation of EHDs' function has also been implicated in human disease (reviewed in [3]). For example, EHD1 is overexpressed in several cancers, which appears to mediate chemotherapy resistance, epithelial-mesenchymal transition and stem cell behavior [23–25]. Mechanistically, EHD1 promotes the trafficking of insulin-like growth factor 1 receptor and other receptor tyrosine kinases to the cell surface, therefore promoting tumorigenesis and metastasis [26]. Mutations in EHD1 are associated with tubular proteinuria and deafness in human [27]. EHD2 knock out mice have increased fat depositions around organs and obese individuals show significantly lowered EHD2 expression levels [19]. Furthermore, overexpression of EHD2 was shown to promote tumorigenesis and metastasis in triple negative breast cancer [28]. Strongly decreased EHD3 expression levels were found in oral squamous cell carcinoma [29] and gliomas [30], while EHD3 overexpression was found in a rat model of ischemic heart failure [22]. EHD4 was shown to regulate urinary water homeostasis [31]. Given the involvement of EHDs in various physiological processes and its dysfunction in pathologies, small molecules that regulate EHDs' activity could be useful tools to study their function in disease and potentially ameliorate some of the disease phenotypes.

Previously, EHD ATPase activity was determined via a High Pressure Liquid Chromatography (HPLC)-based method [4, 7]. However, the HPLC setup and assay parameters are not compatible for high throughput drug screening (HTS). Another study used a Malachite green-based assay for EHD1 and EHD2 [9], but at relatively high protein and substrate concentrations, which prevents direct application in HTS. Various dynamin drug screening efforts have been reported over the last years [32–38]. However, the employed assay parameters cannot be directly applied to EHD proteins due to their weak ATPase activity. Therefore, developing a reproducible ATPase assay of EHDs compatible with HTS is crucial for potential drug screening efforts.

In this study, we have established the Malachite green (MLG)-based assay for measuring the liposome-stimulated ATPase activity of EHD4. MLG is a sensitive and cost-effective agent for determining released inorganic phosphate from ATP hydrolysis, and the assay does not include cumbersome washing or detection steps or the use of radioactive substances [39, 40]. Several assay parameters were optimized to allow HTS in the presence of liposomes. Eventually, we identified and initially characterized a prototypic small-molecule inhibitor for EHD4. The systematic optimization of this screen may be applied to identify inhibitors for other dynamin-related proteins with a low, membrane-stimulated nucleotide hydrolysis activity.

## Materials and methods

### EHD4 expression and purification

An N-terminal truncated construct of mouse EHD4 (Uniprot Q1MWP8, residues 22–541, EHD4$^{\Delta N}$) was expressed as N-terminal His$_6$-fusion followed by a PreScission cleavage site in *E. coli* BL21 DE3 Rosetta (Novagen) from a modified pET28 vector, as described in [7]. Bacterial cultures were grown in Terrific Broth medium (TB) to an OD$_{600}$ of 0.8 at 37˚C. Protein expression was induced at 18˚C by adding 100 μM isopropyl-ß-D-1-thiogalactopyranoside (IPTG) and cultures were grown for another 20 h at 18˚C. Cells were sedimented, resuspended in 50 mM HEPES pH 7.5, 500 mM NaCl, 25 mM imidazole, 2.5 mM β-mercaptoethanol, 2 mM MgCl$_2$ and stored at -20˚C.

Upon thawing, 1 μM DNase (Roche), 500 μM of the protease inhibitor 4-(2-aminoethyl)-benzolsulfonyfluorid hydrochloride (BioChemica) and 5 U/mL benzonase (Merck) were

added (final concentrations). Cells were disrupted by passing them through a microfluidizer (Microfluidics). The lysed bacterial suspension was then centrifuged at 20,000 rpm for 30 min at 4˚C. The supernatant was collected and filtered using 0.45 μm filter packs, and then applied on a Ni$^{2+}$-sepharose column equilibrated with equilibration buffer (50 mM HEPES pH 7.5, 500 mM NaCl, 25 mM imidazole, 2.5 mM β-mercaptoethanol, 2 mM MgCl$_2$). This was followed by a long and short wash step with 50 mM HEPES pH 7.5, 700 mM NaCl, 30 mM imidazole, 2.5 mM β-mercaptoethanol, 1 mM ATP, 10 mM KCl and 2 mM MgCl$_2$ and with equilibration buffer, respectively. Protein was eluted with 20 mM HEPES pH 7.5, 500 mM NaCl, 300 mM imidazole, 2.5 mM β-mercaptoethanol and 2 mM MgCl$_2$, incubated with 1:150 (w/w) His-tagged Human Rhinovirus (HRV) 3C protease and dialyzed overnight in a 10 kDa cutoff membrane against dialysis buffer (20 mM HEPES pH 7.5, 500 mM NaCl, 2.5 mM β-mercaptoethanol and 2 mM MgCl$_2$). To remove the protease and His$_6$-tag, the cleaved protein was applied to a second Ni$^{2+}$ column equilibrated with equilibration buffer to which it bound also in the absence of the His-tag. Protein was eluted with 20 mM HEPES pH 7.5, 500 mM NaCl, 2.5 mM β-mercaptoethanol, 40 mM imidazole and 2 mM MgCl$_2$ buffer. Cleaved protein was concentrated and loaded on Superdex 200 gelfiltration column (GE) equilibrated with size exclusion buffer (20 mM HEPES pH 7.5, 500 mM NaCl, 2.5 mM β-mercaptoethanol and 2 mM MgCl$_2$). Fractions containing pure protein were pooled, concentrated and flash-frozen in liquid nitrogen and stored at -80˚C.

Mouse EHD2 (Uniprot Q8BH64, residues 1–543) and human DNM1L isoform 2 (UniProtID: O00429-3, residues 1–710) were purified in a similar fashion [4, 41].

## Liposome preparation

Different lipid extracts were obtained from Sigma Aldrich with stock concentration of 25 mg/ml in chloroform. For small scale applications, 20 μl from the liposome stock was mixed with 200 μl of 3:1 (v/v) chloroform/methanol mixture in a glass vial. Lipids were dried under a gentle argon stream and stored overnight under vacuum in a desiccator. Dried lipids were hydrated for 5 min by adding 250 μl liposome buffer (20 mM HEPES pH 7.5, 300 mM NaCl) to reach a final concentration of 2 mg/ml and then resuspended by sonication in an ultrasonic water bath for 2 cycles of 30 s each. The freshly prepared liposomes were left for at least half an hour at room temperature before use. Prior to usage, the liposomes were extruded by passing 13 times through a mini-extruder (Avanti) using polycarbonate filters with 0.4 μm pore size. Larger volumes of liposomes were prepared as detailed in S1 Fig.

The following lipids were used in this study: Folch (Sigma-Aldrich, #B1502), brain-derived phosphatidyl serine (PS, Avanti-Polar Lipids, #840032C), synthetic 1-palmitoyl-2-oleoyl-sn-glycero-3-phospho-L-serine (POPS, Avanti-Polar Lipids, #840034P), synthetic 18:1 1,2-dioleoyl-sn-glycero-3-phospho-L-serine (DOPS, Avanti-Polar Lipids 840035).

## Malachite green dye preparation

0.1% MLG dye (Sigma, M9015) was dissolved in 1 M HCl by vortexing. 1% ammonium molybdate (Sigma, 09878-25G) was added and vigorously vortexed until no visible precipitates were observed. The resulting solution was filtered through a 0.45 μm filter and stored in an aluminum foil-wrapped glass bottle in the dark.

## Malachite green ATPase assay

ATPase assays were performed in a 384 well plate at 25˚C (if not otherwise indicated). The assay buffer contained 20 mM HEPES, 150 mM KCl, 0.5 mM MgCl$_2$, pH 7.5. 30 μM of ATP was mixed with 50 μg/ml DOPS in assay buffer. Reaction was started by adding 200 nM

EHD4$^{\Delta N}$. At any given time-point, 25 μl of the reaction was taken out and mixed with 75 μl of the dye in a 384 well plate. The reaction gets quenched by the addition of the dye which denatures the protein. Absorbance at 650 nm was measured in a plate reader (Tecan, Infinite M200).

Turnover of substrate was limited to 10% for Michaelis-Menten constant determination ($K_m$) and was calculated by fitting initial velocities to a non-linear least fit squares to the Michaelis Menten equation ($v_0 = v_{max}[S]/(K_m+[S])$) using GraphPad Prism 7.05. ATPase assay and $K_m$ determination were done in duplicates and triplicates, respectively.

## HPLC-based ATPase assay

ATPase assay was performed as mentioned above and at defined time points, reaction aliquots were diluted three times in reaction buffer (50 mM Tris, 150 mM NaCl, pH 7.5) and quickly flash frozen in liquid nitrogen. Samples were applied to an HPLC and nucleotides were separated with reversed phase column (C18 100*4.6 mm) equilibrated with running buffer containing 100 mM potassium phosphate buffer pH 6.5, 10 mM tetrabutylammonium bromide, 7.5% (*v/v*) acetonitrile. Adenine nucleotides were detected by absorption at 254 nm and quantified by integrating the corresponding nucleotide peaks and determining the ADP/ATP ratios.

## Drug screening and IC$_{50}$ determination

Screening was conducted at the Screening Unit of the Leibniz-Forschungsinstitut für Molekulare Pharmakologie (FMP) using a kinase drug library and the 'diversity set'. 15,840 compounds were tested as singlets in 384 well plate format. A master mix (MM) was prepared in the assay buffer to have a final ATP concentration of 30 μM and DOPS concentration of 50 μg/ml. A 10 mM DMSO solution of the compounds was serially diluted, first with DMSO and then with the MM to a final concentration of 10 μM (1% of DMSO), using the Tecan-Evo workstation (pipetting robot). EHD4$^{\Delta N}$ (200 nM) was pipetted using multi-drop dispensing cassette. To have homogenous enzymatic activity, the plates were vortexed using a microplate mixer at 2,000 rpm for 10 s. The plates were incubated for 15 min at room temperature and 75 μl of MLG dye added with the cassette. The plates were measured in a Perkin Elmer Envision plate reader.

Data were analyzed with an automated pipeline [42] using KNIME software [43]. For the individual plates, the corresponding Z′ values and SNR were calculated. Compared to the control′s absorbance at 650 nm (Abs650), compounds were classified as inhibitors if the %Abs650 was ≤ 25%. These hits were picked from the drug library and used for further validation of IC$_{50}$ determination and counter screening. IC$_{50}$ values were determined in the same setup with varying inhibitor concentrations from 0.125 μM to 50 μM.

## Thermal shift assay (TSA)

TSA was done in a CFX96 touch-real time PCR using a fluorescent dye from ThermoFisher Scientific (catalogue number 4461146). Reactions (5 μM EHD4$^{\Delta N}$, 50 μM inhibitor and 1X dye) were prepared on ice, to a final volume of 50 μl and transferred to a 96 well PCR plate and then sealed with a plastic film. CFX96 was precooled to 4˚C and the PCR plate was inserted. The fluorescence was measured from 4˚C to 80˚C with 0.5˚C temperature steps. Protein melting temperature was calculated with the provided software (Bio-Rad) and plotted as bar diagram using GraphPad Prism 7.05.

## Compounds

The identified EHD4 inhibitors are N-(2-hydroxyphenyl)-6-(3-oxo-2,3-dihydro-1,2-ben-zothiazol-2-yl)hexanamide (MS1, Molport, 008-333-699), N-(4-methyl-1,3-thiazol-2-yl)-3-phenoxybenzamide (MS8, Enamine, Z30820224) and 6-(Morpholino(4-(Trifluormethyl) phenyl)methyl)-1,3-Benzodioxol-5-ol (MS10, Enamine, Z99601030).

## Results

### Developing an EHD4 ATPase assay suitable for drug screening

We aimed for developing a prototypic small molecule inhibitor for the EHD family. From the four mammalian EHDs, only EHD2 and EHD4 can be bacterially expressed and purified in quantities compatible with HTS (S2A Fig) [4, 7]. EHD2 has a very low basal ATPase activity ($k_{cat}$ ~1 h$^{-1}$) which can only be moderately stimulated in the presence of lipids ($k_{cat}$~ 0.1 min$^{-1}$) [4]. This low activity is not suited for reliable HTS. Instead, we chose an EHD4 construct with a truncated N-terminus (EHD4$^{\Delta N}$), which was previously shown to have a higher basal and stimulated ATPase activity compared to EHD2 [7]. Since the basal EHD4$^{\Delta N}$ activity is still low, we focused on its liposome-stimulated ATPase activity.

To apply an MLG-based assay, the maximum absorption wavelength for the MLG dye in complex with orthophosphate (colorimetric complex) was initially determined. Various absorption maxima for MLG have been previously reported [44–47]. The dye used in this study showed an absorption maximum between 640 nm and 650 nm (S2B Fig). Orthophosphate detection was linear up to 100 μM phosphate (S2C Fig).

We next analysed interference of the MLG assay with assay components. ATP yielded a moderate background signal, maybe from phosphate contaminations, thereby restricting the maximal ATP concentration for screening (S2D and S2E Fig). Furthermore, previously reported ATPase activities of EHD4$^{\Delta N}$ [4, 7] were recorded in the presence of Folch liposomes, which resulted in a large background signal in the MLG assay (Fig 1A). In contrast, liposomes prepared from phosphatidyl serine (PS) from natural sources and synthetic 1,2-dioleoyl-sn-glycerol-3-phospho-L-serine (DOPS) gave a negligible background signal (Fig 1A). However, only with synthetic DOPS, the EHD4$^{\Delta N}$ enzymatic activity was reproducibly stimulated (S2F and S2G Fig), while the non-stimulated ATPase was hardly above background (S2H Fig). EHD4$^{\Delta N}$ enzymatic activity was higher at 30°C and 25°C compared to 37°C (Fig 1B and S2I Fig). For easier application in the HTS assay, we used room temperature from hereon (25°C).

Potassium ions can influence the ATPase activity of various ATPases [48]. For dynamin and related enzymes, potassium has a catalytic role by binding into the active site next to the β-phosphate during GTP hydrolysis [49, 50]. We therefore tested the influence of potassium ions on EHD4$^{\Delta N}$ ATPases activity in addition to 150 mM NaCl already present in the assay buffer. In this case, 150 mM KCl resulted in the maximal activity (Fig 1C). Replacing all sodium ions in the assay buffer with potassium led to a slight further increase in the ATPase activity (Fig 1C). The optimized buffer conditions were therefore 20 mM HEPES (pH 7.5), 150 mM KCl and 0.5 mM MgCl$_2$.

EHD4$^{\Delta N}$ ATPase assays were previously performed at μM protein concentrations which is too high for HTS carried out at typical inhibitor concentration of 10 μM. Therefore, we performed EHD4$^{\Delta N}$ titrations and found that EHD4$^{\Delta N}$ was active also at high nM concentrations with the improved assay conditions (Fig 1D). Similarly, to analyze the liposome concentration dependency, we titrated different DOPS concentration at 400 nM EHD4$^{\Delta N}$ and found maximal activity at DOPS concentration of 50 μg/ml, e.g. 20-fold lower than previously described (Fig 1E) [4, 7]. As protein and liposome concentrations mutually affect the enzymatic activity,

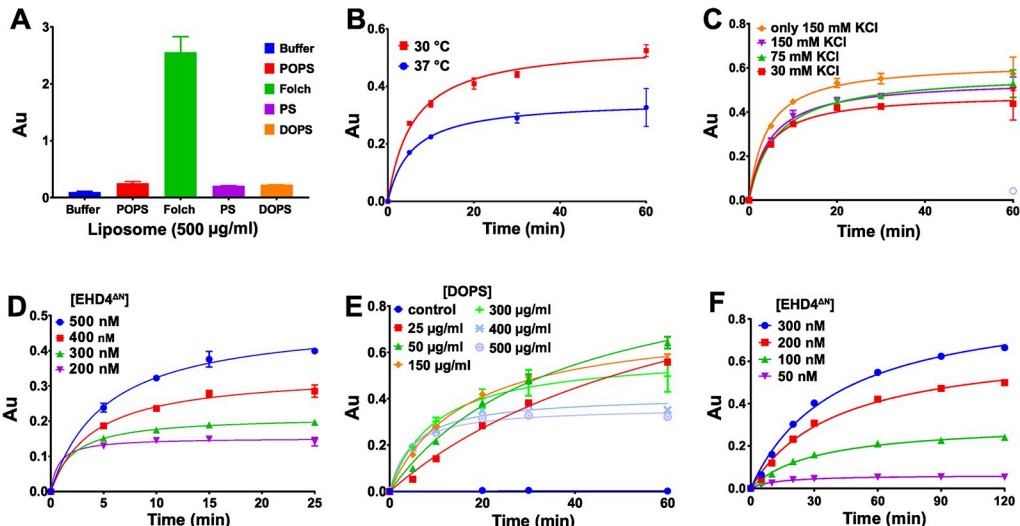

**Fig 1. Optimization of the Malachite-Green ATPase assay for EHD4$^{\Delta N}$. A** Background signal in the MLG assay arising from different lipids: Synthetic POPS, Folch, PS isolated from porcine brain and synthetic DOPS at 500 μg/ml. AU–arbitrary unit. **B** MLG-based ATPase assays for 4 μM EHD4$^{\Delta N}$, 500 μg/ml DOPS and 50 μM ATP were done at 37˚C and 30˚C. A further decrease to 25˚C did not affect the enzymatic activity compared to 30˚C (S2I Fig). **C** MLG-based ATPase assays at 1 μM EHD4$^{\Delta N}$, 500 μg/ml synthetic DOPS and 30 μM ATP done at 25˚C in the presence of different concentrations of potassium ions. Removal of sodium ions in the presence of 150 mM KCl (denoted as 'only 150 mM') resulted in maximum ATPase activity. **D** EHD4$^{\Delta N}$ ATPase activity in the presence of 500 μg/ml DOPS and 30 μM ATP at 25˚C was screened at the indicated protein concentrations. **E** Dependency of the EHD4$^{\Delta N}$ ATPase activity on synthetic DOPS-liposome concentration was analyzed at 400 nM EHD4$^{\Delta N}$, 50 μM ATP in standard buffer at 25˚C. At lower DOPS-liposome concentration, an increase in enzymatic activity was observed. 50 μg/ml synthetic DOPS-liposomes were used in all subsequent assays. **F** Different EHD4$^{\Delta N}$ concentrations were tested in the presence of 50 μg/ml DOPS liposomes and 50 μM ATP at 25˚C. 200 nM EHD4$^{\Delta N}$ was chosen to have a SNR of at least 2. Data points represent the mean of two or more independent experiments and the error bar signifies the range or standard deviation of the measurements. When the range/standard deviation is smaller than the size of the data points, it is not displayed.

we re-analyzed the ATPase dependence on EHD4$^{\Delta N}$ concentrations at a constant DOPS concentration of 50 μg/ml. A protein-dependent increase in ATPase activity was observed (Fig 1F). To obtain a signal to noise ratio (SNR) of at least 2 while at the same time using a low protein concentration, 200 nM EHD4$^{\Delta N}$ and 50 μg/ml DOPS were chosen for the subsequent enzymatic assays (S2J Fig).

## Enzyme kinetics and assay quality

For drug screening, we aimed at applying a substrate concentration close to the Michaelis-Menten constant ($K_m$) to target competitive and non-competitive inhibitors [51]. The apparent $K_m$ of EHD4$^{\Delta N}$ at a protein concentration of 1 μM was (20 ± 3) μM with a $k_{cat}$ of 2.5 min$^{-1}$ (Fig 2A and 2B). However, due to the slightly higher SNR of 2.1 vs. 1.6 at 30 μM vs. 20 μM ATP after 15 min, respectively (Fig 2C), we chose 30 μM ATP as substrate concentration for the subsequent HTS screens. The final assay parameters were 200 nM EHD4$^{\Delta N}$, 30 μM ATP, 50 μg/ml DOPS at 25˚C in 20 mM HEPES (pH 7.5), 150 mM KCl, 0.5 mM MgCl$_2$ (Fig 2D), which resulted in an excellent quality indicator for HTS, a Z´-factor [52], of 0.885.

To visually assess data reproducibility of our assay, we performed a pilot screen using three plates from the drug library, measured in two technical replicates. Data were analyzed using the Bland-Altman method, a statistical technique used to assess the agreement between two different measurement methods or to calculate a repeatability coefficient [53]. To achieve the latter, the difference of percentage activities of the two technical replicates were plotted against

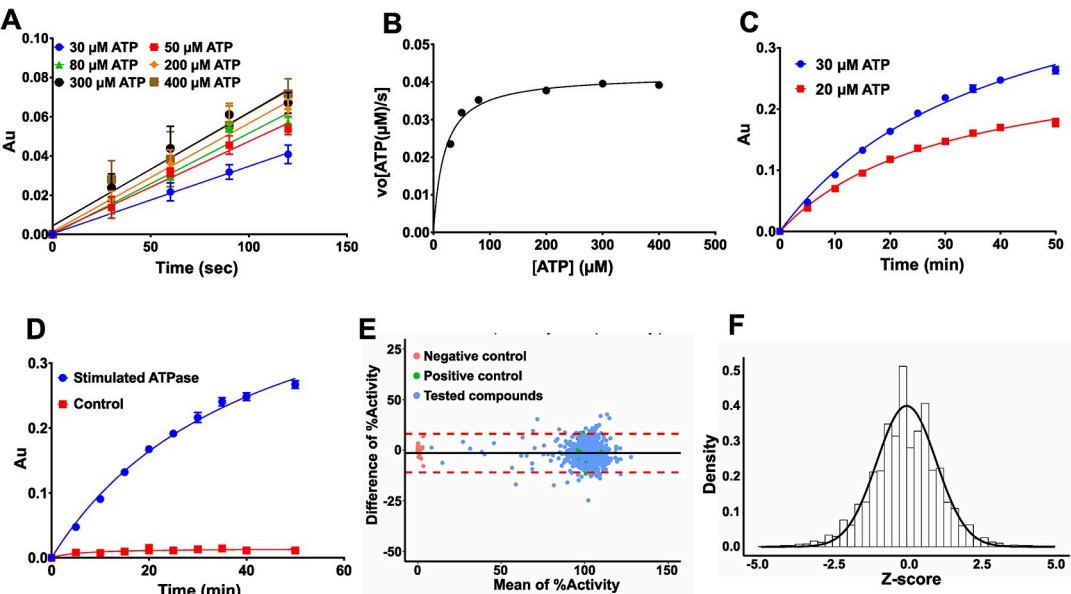

**Fig 2. Reaction kinetics and assay quality. A** Initial velocity of 1 μM EHD4$^{\Delta N}$ at different ATP concentrations in the presence of 50 μg/ml DOPS-liposomes were determined by calculating the slope of the linear reaction. **B** K$_m$ was determined by first calculating the amount of hydrolyzed ATP in (**A**) using the standard curve mentioned in S2C Fig. and then plotting the initial rates of the reactions versus the substrate concentration. The kinetic parameters for EHD4$^{\Delta N}$ are K$_m$ = (20 ± 3) μM, k$_{cat}$ = (2.51 ± 0.07) 1/min, v$_{max}$ = 2.51 μmoles ATP/min. **C** ATPase activity 200 nM of EHD4$^{\Delta N}$ in the presence of 50 μg/ml liposomes at 20 and 30 μM ATP concentration. The signal to noise ratio was 1.6 for 20 μM ATP whereas it was 2.1 with 30 μM ATP at 15 min. Therefore, 30 μM ATP was chosen for drug screening. **D** The final optimized assay conditions were 200 nM EHD4$^{\Delta N}$, 50 μg/ml DOPS and 30 μM ATP at 25˚C. Control represents assay without EHD4$^{\Delta N}$. Under the optimized assay conditions, the ATPase activity is almost linear until 15 min, the incubation time used for drug screening. **E** Bland-Altman plot from pilot screen using 3 library plates in technical replicates. Plot shows differences of repeated measurements falling into a 95% confidence interval of ± 9.5%, suggesting high confidence between replicates. The mean Z´-factor of the pilot screen was 0.885. **F** From the same pilot screening, a Z-score histogram was plotted showing normal distribution. Data points represent the mean of two or more independent experiments and the error bar signifies the range or standard deviation. When the range/standard deviation is smaller than the size of the data point, it is not displayed.

their mean (Fig 2E). The coefficient of repeatability was then calculated from the observed differences and showed a high confidence between the replicates, with the differences of repeated measurements falling into a 95% confidence interval of ±9.5%. In a Z-score histogram (Fig 2F), Z-scores were equally distributed around the zero midpoint, with a shape resembling a normal distribution, indicating that data normalization was effective in removing the batch-to-batch variation in absolute signals. Z´-factors of the plates were between 0.75–0.9. Finally, no effect of 1.5% DMSO on the enzymatic activity was observed (S2K Fig). This rendered the assay compatible for the HTS setup, since all small molecules in the HTS were diluted from DMSO stock solutions, yielding a final DMSO concentration in the assay of 1%.

## Drug screening

Using the optimized assay conditions, we screened ~16,000 compounds from the drug library of Leibniz-Forschungsinstitut für Molekulare Pharmakologie (FMP) for EHD4$^{\Delta N}$ ATPase inhibition in single replicates using a single time point of 15 min. Data were analyzed with an automated pipeline [42]. Compounds showing a decrease in enzymatic activity by 25% were grouped as inhibitors. Primary hits were validated in duplicates and counter-screened for false positives. For counter screening, we tested 30 μM phosphate in assay buffer against 10 μM inhibitor in the MLG assay to determine if inhibitors interfered with the colorimetric reaction.

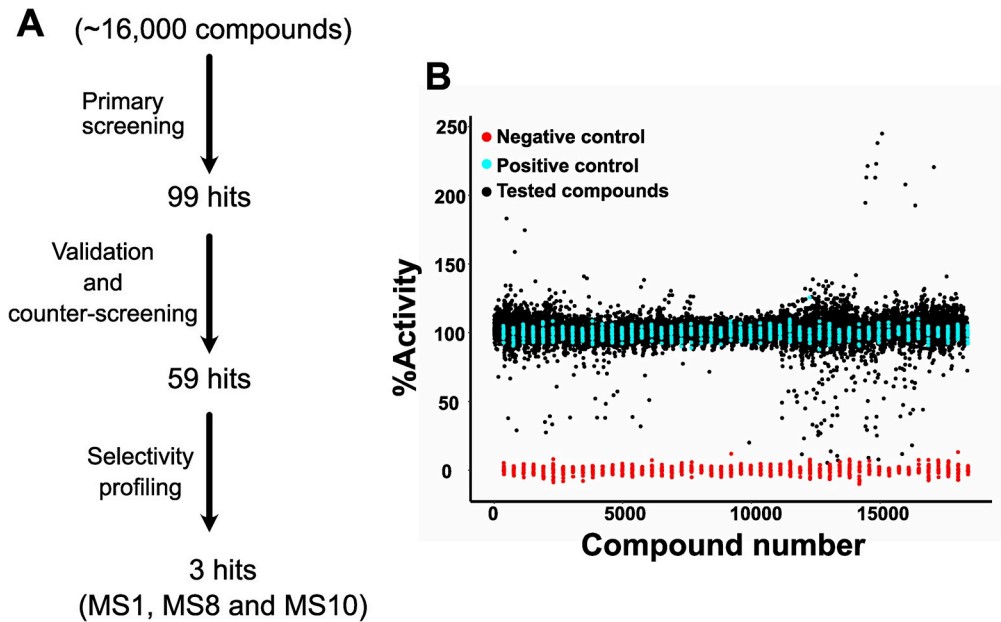

**Fig 3. EHD4 drug screening. A** Schematic representation of EHD4 drug screening. Approximately 16,000 compounds were tested at 10 μM concentration in a primary screen against EHD4$^{\Delta N}$ with the MLG assay. Initial hits were validated in duplicates and counter-screened to eliminate false positives, e.g. compounds that interfered with the colorimetric reaction. The final three compounds as primary hits from drug screening were selected based on their IC$_{50}$ value and chemical structure. Thus, compound with an unstable chemical structure prone to oxidation or degradation, with a tendency to form covalent bonds, and containing a PAINS substructure were eliminated, as well as compounds showing a steep fall in IC$_{50}$ curves (see Discussion). **B** Scatter plot of the single point screening data. Compound numbers are for visualization purposes. Cyan represents the positive controls without compounds, red the negative controls without EHD4$^{\Delta N}$, and black the tested compounds. Potential activators, e.g. compounds apparently showing a higher activity, were all later found to be false positives in a counter screening experiment by inducing themselves a signal in the MLG assay. The mean Z´-factor for the complete drug screening, including the data from pilot screen, was 0.827.

At this high phosphate concentration, even subtle interference of the inhibitors with the colorimetric assay can be detected. IC$_{50}$ values for the validated hits were determined. An overview of the drug screening and a scatter plot of the single point screening data is shown in Fig 3A and 3B.

For the final compound selection, we considered IC$_{50}$ values and excluded chemical structure of compounds that were unstable and hence prone to degradation. Further selection criteria are outlined in the discussion. We proceeded with three primary hits, MS1, MS8 and MS10, showing IC$_{50}$ values of 0.92 μM, 3.8 μM and 2.9 μM respectively (Fig 4), with maximal inhibition of 60–80%.

## Biochemical characterization of the inhibitors

We re-purchased the three compounds and validated them first via the MLG assay (Fig 5A) and then via HPLC to remove any assay bias (Fig 5B). All three compounds inhibited EHD4$^{\Delta N}$ enzymatic activity in both assays. IC$_{50}$ measurement for the purchased compounds (S3 Fig) were similar to the compounds picked from the drug library (Fig 4). To examine specificity, we investigated the inhibitors against EHD2. Due to the exceptionally low ATPase activity of EHD2 and the high background signal by Folch lipids required for its ATPase stimulation, detection in the MLG assay was not possible. Thus, EHD2 ATPase assay was instead measured by HPLC (S4A Fig) [4]. MS1 and MS10 inhibited EHD2 enzymatic activity whereas MS8 had

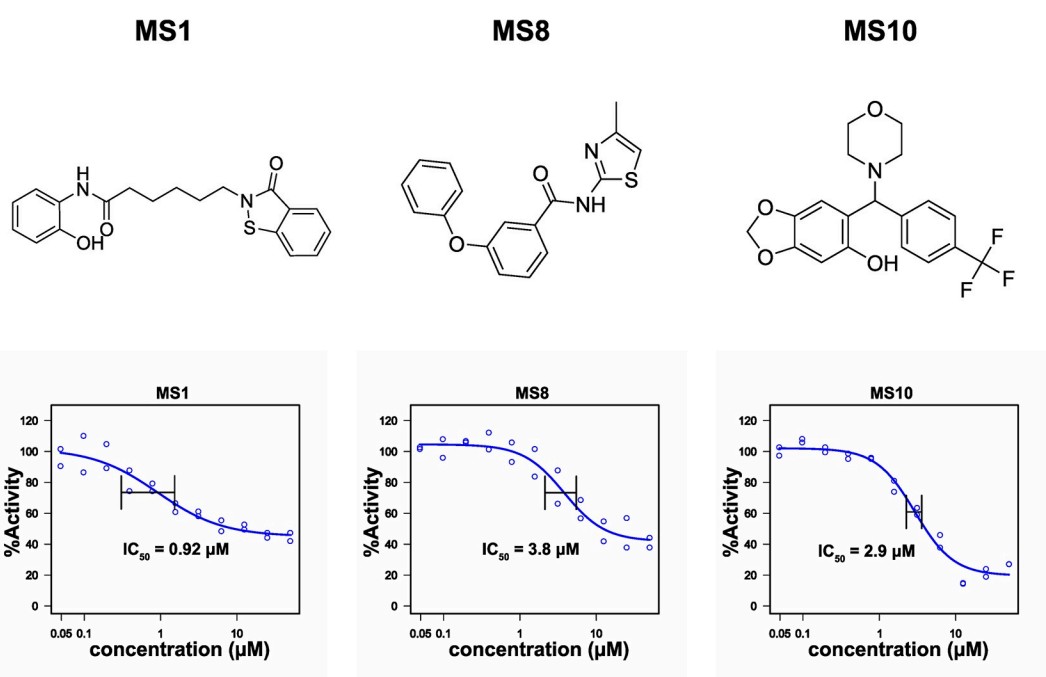

**Fig 4. Primary hits from drug screening.** Chemical structure and $IC_{50}$ values for the primary hits from the EHD4 drug screen. Hits were picked from the drug library and $IC_{50}$ values were determined in duplicates against an inhibitor concentration range from 0.125 μM to 50 μM.

no effect (Fig 5C). In thermal shift assay, EHD4$^{\Delta N}$ showed a melting temperature of 44.5˚C (S2L Fig). MS1 had a destabilizing effect on EHD4$^{\Delta N}$, as evident by a negative $\Delta T_m$ of -5˚C. MS10 also diminished EHD4$^{\Delta N}$ stability whereas MS8 had no such effect (Fig 5D, S2M Fig). All three compounds were further counter-screened against a distantly related dynamin family member, DNM1L, by the MLG assay (S4B–S4D Fig). None of them showed an effect on the stimulated GTPase activity of DNM1L (Fig 5E, S4E Fig). This further validated the specific inhibition of MS8 on EHD4 and excluded an effect of the inhibitors on liposome integrity, required for the activation of DNM1L.

## Structure activity relationship

Based on the chemical structure of the hit MS8, a substructure search of the FMP library was conducted. Two possible dissections of chemical bonds, either the amide or diphenylether bond, have been used in this search. The resulting fragments were analyzed according to the number of compounds inside the FMP library with highest diversity of the chemical space and the optimal coverage of chemical vectors to gain a structure activity relationship. In this process, the N-(thiazol-2-yl)benzamides substructure (Fig 5F) yielded the best results of a suitable compound set containing this fragment.

Preliminary SAR data were derived by commercially available 55 compounds from the substructure search of N-(thiazol-2-yl)benzamides in the FMP library with a broad spectrum of substitution pattern. Only three compounds (Z5, Z7 and Z8, see S5 Fig) showed 20% or more inhibition of EHD4$^{\Delta N}$ ATPase activity in the MLG assay at 50 μM inhibitor concentration and were classified as active, whereas the remaining were classified as inactive (see S1 Table).

Analysis of the SAR substructures showed that a connection between the benzamides to the thiazole rings in 2-position is mandatory for the activity. The substitution in 3-position of the

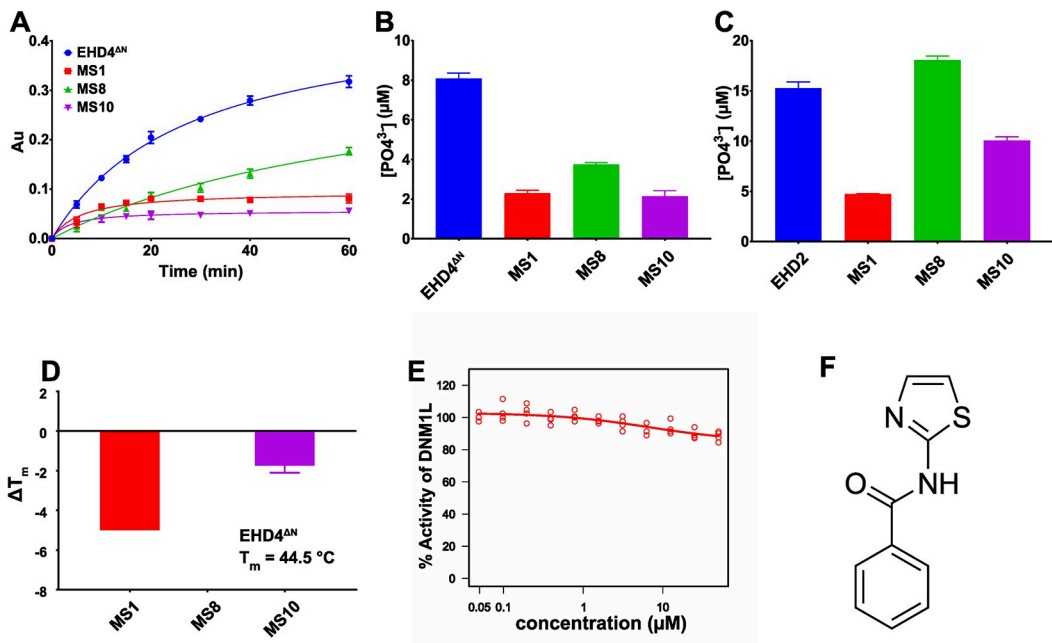

**Fig 5. Biochemical characterization of the primary hits. A** One-hour time course of the MLG-based enzymatic ATPase assay in the presence of 50 μM inhibitor. EHD4$^{\Delta N}$ (blue curve) represents the positive control. **B** Primary hits were validated at 50 μM inhibitor concentration in an HPLC setup to exclude assay bias. ATPase assays done in **A** and **B** were in standard conditions. **C** Primary hits were tested against EHD2. Only MS8 had no effect on EHD2 ATPase activity. ATPase assay was done at an EHD2 concentration of 5 μM in the presence of 500 μg/ml Folch liposomes and 50 μM ATP in assay buffer at 30°C. **D** Changes in melting temperature of 5 μM EHD4$^{\Delta N}$ were studied in the presence of 50 μM inhibitor by thermal shift assay. MS1 and MS10 destabilized the protein, as evident by negative $\Delta T_m$, whereas MS8 had no effect on the stability of EHD4$^{\Delta N}$. **E** MS8 did not inhibit the GTPase activity of DNM1L. Shown is an IC$_{50}$ curve plotted with an inhibitor concentration range from 0.125 μM to 50 μM. **F** MS8 substructure obtained by dissecting the diphenlyether bond. This substructure was used in our chemical search to have the highest diversity of the chemical space and the optimal coverage of chemical vectors to gain a structure-activity relationship.Data points represents the mean of two or more independent experiments and the error bar signifies the range or standard deviation. When the range/standard deviation is smaller than the size of the data point, it is not displayed.

thiazole ring can be either alkyl chains or aromatic sided chains (see S5 Fig and S1 Table, MS8, Z5 and Z7). Substitution in the 5-position diminishes the activity significantly (see S1 Table, e.g. compounds 106268 or 106333). Any annealed rings on the benzamide or between the 4 and 5 position of the thiazole ring abandoned the activity completely (see S1 Table, e.g. compounds 105229, 201598 or 203985 and many more). The benzamide ring does not tolerate substitutions in *ortho*-position (S1 Table, compounds 209726 or 301620). Activity is gained via substitution in *meta*-position with methoxy or phenoxy rings only (S5 Fig and S1 Table, MS8, Z5, Z7 and Z8). Any other substitution with smaller functional groups in *para* and/or *meta*-position seems to diminish the activity (S1 Table, e.g. compounds 110741, 100723 or 209726), but we have only a limited number of examples. In the future, efforts will be made to re-purchase commercial available N-(thiazol-2-yl)benzamides based on the knowledge gained in the summarized SAR (Fig 6) in order to further boost the inhibitory activity of EHD4.

## Discussion

There are no chemical inhibitors for EHD proteins described to date. This study reports the identification of an EHD4 inhibitor by establishing a robust, reproducible EHD4 ATPase assay in the presence of liposomes, which is compatible with drug screening. Several problems had

**Fig 6. Initial SAR data on commercially available compounds tested in the EHD4 assay based on N-(thiazol-2-yl) benzamides as the central core scaffold.** Substructure search resulted in 55 commercially available compounds which were picked from the drug library and re-tested. Only three of them inhibited EHD4 enzymatic activity by 20% or more (S5 Fig).

to be overcome during optimization of the MLG-based assay. The basal ATPase activity of EHD4$^{\Delta N}$ is hardly above the background in the MLG-based assay and could therefore not be used for inhibitor screening. In contrast, the higher basal GTPase activity of dynamin in conjunction with a competitive and highly sensitive fluorescence polarization immunoassay has been successfully used for HTS [38]. Furthermore, the stimulated ATPase activity of EHD4$^{\Delta N}$ was not reproducible with liposomes derived from natural extracts, which inevitably vary from batch to batch. Synthetic DOPS (18:1) employed in this study has similar properties to naturally occurring brain PS, but it is more stable towards oxidation, which makes it a suitable substitute. DOPS has been used in drug discovery studies for other purposes, for example as a component of lipid mixtures to mimic the plasma membrane [54], in liposome preparations for anti-tumor applications [55] and amyloidosis studies [56]. Unlike dynamin's stimulated GTPase activity [40], ATP hydrolysis of EHD4$^{\Delta N}$ plateaued after 5 min in the initial conditions. However, optimization of the protein and lipid concentrations in our assay led to an almost linear stimulated ATPase activity for EHD4$^{\Delta N}$, rendering it compatible with determination of Michaelis-Menten parameters (Fig 2A and 2B) and our drug screening efforts.

The stimulated ATPase activity of EHD4$^{\Delta N}$ was higher at 30°C versus 37°C, which may relate to different fluidity of the lipid bilayer or to a decreased stability of EHD4$^{\Delta N}$ in the presence of liposomes at higher temperatures in the *in vitro* assay. Finally, we aimed to exclude inhibitors that indirectly act on EHD4's stimulated ATPase activity by affecting liposome integrity. Accordingly, some dynamin inhibitors were previously shown to interfere indirectly with dynamin's function [57], possibly by interacting with lipids and detergents [37, 58]. To reduce the chance of such a scenario, we counter-screened the inhibitors against the stimulated GTPase activity of DNM1L, which also requires intact liposomes.

Utilizing the MLG-based assay, we screened ≈16,000 compounds for their ability to inhibit EHD4$^{\Delta N}$ ATPase activity. After validating and counter screening, we excluded unstable chemical structure of compounds prone to degradation and or oxidation and potential covalent bonds forming compounds (Michael acceptor) that could lead to non-specific cellular binding [59]. We also removed substances inducing a sudden concentration-dependent activity drop of EHD4$^{\Delta N}$, which may hint at the compound forming aggregates and sequestering the enzyme [60, 61]. Furthermore, we applied substructure filtering for identifying Pan Assay Interference Compounds (PAINS) [62] and removed compounds showing high number of reported biological activities in the ChEMBL database [63]. In this way, we limited the further analysis to three compounds MS1, MS8 and MS10, which inhibited EHD4$^{\Delta N}$ with IC$_{50}$ values

in the low micromolar range. The preference for MS8 was guided by structural considerations and specificity towards the target. Although not automatically excluded via PAINS filter, MS1 consists of an isothiazolinone substructure which was listed as a nuisance compound in a GSK panel list [64]. Additionally, it is described that the structure might degrade over time via oxidation to sulfoxide and sulfone following by ring-opening [65]. MS10 might be synthesized via a mannich-like condensation mechanism using the corresponding ketone and the morpholine to the imine followed by reductive amination. One could consider a retro-condensation mechanism which would lead to degradation of the MS10. In comparison to MS1 and MS10, the structure of MS8 is very stable consisting of two aromatic groups connected via an amide bond. Moreover, MS8 did not inhibit enzymatic activity of the closely related EHD2 and also had no effect on the enzymatic activity of another dynamin superfamily protein member, DNM1L.

The exact inhibition mechanism of the compounds requires further analyses. Substances directly interfering with ATP binding, membrane binding or oligomerization would be expected to reduce stimulated ATP hydrolysis reaction. Also, allosteric inhibition of the ATPase could be envisaged. Furthermore, compounds stabilizing the EHD oligomers may lead to interference of catalysis, e.g. by blocking ADP release after one round of hydrolysis. A structure of an EHD4-inhibitor complex may provide details of the inhibition mode, but so far, we were not able to obtain a crystal structure of such a complex.

MS8 could be used to better understand the EHD4-mediated signaling pathways. EHD4 was discovered as a protein induced by nerve growth factor (NGF) and mediates cytoplasmic signaling of NGF through its receptor, TrkA (Tropomyosin receptor kinase A) [21]. Other roles of EHD4 include internalization of the trans-membrane protein Nogo-A into neuronal cells by EHD4-mediated endocytosis. Nogo-A is expressed in the adult central nervous system (CNS), where it inhibits axonal growth, regeneration and plasticity [66]. EHD4 along with EHD1 is also involved in the endocytosis of L1/neuron-glia cell adhesion molecule (NgCAM) in neurons known to regulate axonal growth [67]. Although progress has been made in addressing the specific role of EHD4 in this membrane trafficking pathway, several open questions remain. For example, it is unclear whether single or multiple routes for EHD4-mediated membrane trafficking exist [68]. Furthermore, whether other EHD proteins are also involved in co-regulating endocytosis for TrkA still needs to be addressed. MS8 may help to dissect EHD4's cellular function. In particular, it will be interesting to explore whether MS8 shows an effect on endosomal signaling in neurons. If such an effect would be observed, even possible therapeutic applications could be considered. For example, a role for a more potent MS8 derivative could be to suppress Nogo-A signaling, which may help in functional recovery of the adult CNS in the aftermath of an injury by stimulating regeneration and nerve fiber growth. Such function would be supported by previous studies showing that EHD4 ATPase-deficient mutant completely blocks the internalization of Nogo-A [66].

In future studies, a combined approach of HTS and structure-guided drug design could result in more potent and highly selective EHD4 inhibitor. Our initial SAR studies have indicated preferred chemical modification of MS8 in order to develop such improved inhibitors.

## Supporting information

**S1 Fig. Schematic representation of liposome preparation with the help of a rotary evaporator. A** Overview of the experimental setup. Nitrogen gas instead of vacuum was used to evaporate the chloroform/methanol mix from the lipid solution. For this, the vacuum port was removed and a 50 ml serological pipette attached to a nitrogen gas tubing into the glass vial (VWR 548–0156). 500 μl of DOPS (Avanti Polar Lipids) were mixed with 6 ml of a

chloroform/methanol mixture (3:1 v/v) in the previously mentioned glass vial. This vial was attached to a Rotavapor R-300 (Buchi) with the bottom part of the vial dipping into the heating bath kept at 25˚C. A gentle nitrogen stream was introduced into the vial and adapted to not observe ripples on the surface of the solution. The rotation speed was set to 175 rpm. **B** Approximately 25 min later, a regular lipid monolayer formed on the glass vial after removal of chloroform/methanol mixture using the Rota vapor. **C** Same lipids after overnight (O/N) drying.
(TIF)

**S2 Fig. EHD4$^{\Delta N}$ ATPase assay optimization. A** Coomassie-stained SDS-PAGE gel showing fractions of the final gel filtration peak of the EHD4$^{\Delta N}$ purification. **B** Absorption maxima of the colorimetric complex formed between the MLG dye, molybdate and orthophosphate. AU–arbitrary units. **C** Standard curve of orthophosphate in the MLG assay. The determined fit parameters of the curve are shown on the top left. **D** Background signal from ATP possibly derived from phosphate contaminations is directly proportional to the ATP concentration in the assay. **E** ATP hydrolysis in the absence of EHD4$^{\Delta N}$ and DOPS in the acidic environment of the MLG dye over time. Note that higher ATP concentration leads to a higher background signal. **F** ATPase activity of EHD4$^{\Delta N}$ at 25˚C at 2 μM EHD4$^{\Delta N}$, 200 μM ATP and 300 μg/ml liposomes composed of natural PS in assay buffer was detected by the MLG assay, but it was not reproducible. R1-R5 represents different repetitions of the EHD4 assay under supposedly identical conditions. **G** ATPase assay of EHD4$^{\Delta N}$ at 4 μM EHD4$^{\Delta N}$, 30 μM ATP and 500 μg/ml synthetic DOPS liposomes in assay buffer were conducted at 25˚C. R1, R2 and R3 represent three independent preparations of DOPS liposomes from different batches, resulting in similar activities. **H** MLG-based ATPase assay at 4 μM EHD4$^{\Delta N}$, 30 μM ATP at 25˚C in the absence (non-stimulated) or presence of 500 μg/ml DOPS (stimulated). Control represents the assay without EHD4$^{\Delta N}$. **I** ATPase assay comparison at 30˚C and at 25˚C at a concentration of 20 μM ATP and 4 μM EHD4$^{\Delta N}$ and 500 μg/ml DOPS liposomes in assay buffer. **J** MLG-based ATPase assay at 200 nM EHD4$^{\Delta N}$, 50 μg/ml DOPS and 30 μM ATP at 25˚C without background subtraction. Signal:noise is plotted on the right y-axis. Control represents the assay without EHD4$^{\Delta N}$. **K** Inclusions of 1.5% DMSO had no effect on EHD4$^{\Delta N}$ enzymatic activity in the presence of liposomes, rendering the assay compatible with the HTS setup. **L** Thermal shift assay of EHD4$^{\Delta N}$ at a protein concentration of 5 μM to determine the melting temperature of the protein. **M** Thermal shift assay of 5 μM EHD4$^{\Delta N}$ with 50 μM each of MS1, M8 and MS10 to determine the melting temperature of the protein in presence of the inhibitors. Except for **B** and **F**, data points represent the mean of two or more independent experiments and the error bar signifies the range or standard deviation. When the range/standard deviation is smaller than the size of the data point, it is not displayed.
(TIF)

**S3 Fig. IC$_{50}$ for the purchased primary hits.** IC$_{50}$ values for the purchased primary hits were similar to the previously determined IC$_{50}$ values of the compounds from the drug library. Repurchased MS1 showed an IC$_{50}$ value of 2.4 μM (vs 0.92 μM with the compound from the drug library), MS8 an IC$_{50}$ value of 4.9 μM (vs 3.8 μM with the compound from the drug library), MS10 an IC$_{50}$ value of 4.8 μM (vs 2.9 μM with the compound from the drug library).
(TIF)

**S4 Fig. DNM1L and EHD2 enzymatic assay. A** EHD2 ATPase activity by an HPLC-based method. Assay conditions were 5 μM EHD2, 50 μM ATP and 1 mg/ml Folch liposomes at 30˚C. Folch liposomes were used here as the HPLC-based assay is compatible with Folch liposomes. **B** Initial velocities of DNM1L GTPase activity at 300 nM DNM1L, and, 500 μg/ml

synthetic DOPS, T = 25˚C and at different GTP concentrations were determined by a linear fit. **C** $K_m$ was determined by first calculating the amount of hydrolyzed GTP in (**B**) using the standard curve reported in S2C Fig and then plotting the initial rates of the reactions versus the substrate concentration. The kinetic parameters for DNM1L are $K_m$ = (66 ± 9) μM, $k_{cat}$ = (3.81 ± 0.15) 1/min, $v_{max}$ = 1.15 μmoles GTP/min. **D** GTPase assay monitoring DNM1L activity were done with the final optimized parameters, which were 300 nM DNM1L, 40 μM GTP, 200 μg/ml synthetic DOPS liposomes, T = 25˚C in 20 mM HEPES (pH 7.5), 150 mM KCl, 0.5 mM $MgCl_2$. Z´ and SNR were 0.72 and 2.1 at 20 min for the stimulated GTPase activity. **E** One-hour time course of the MLG-based enzymatic GTPase assay in the presence of 10 μM inhibitor. DNM1L (blue curve) represents the positive control. The data points except in **C** represent the mean of two independent experiments and the error bar signifies the range of the fit. When the range is smaller than the size of the data point, the error bar is not displayed. (TIF)

**S5 Fig. Compound structure of the Z5, Z7 and Z8.** The three compounds that showed inhibition towards EHD4$^{\Delta N}$ at 50 μM inhibitor concentration by 20% or more in the MLG assay, among the 55 compounds from the substructure search in our SAR study. Z5, Z7 and Z8 inhibited EHD4$^{\Delta N}$ enzymatic activity by 21%, 44% and 38% respectively. The N-(thiazol-2-yl) benzamide moiety, used for our chemical search is encircled in red. (TIF)

**S1 Table. 55 compounds containing the MS8 substructure highlighted in red.** (PDF)

**S1 Raw images.** (PDF)

## Acknowledgments

We thank Dr. Marc Nazaré, Dr. Peter Lindermann and Jerome Paul for usage of and support with their Rotavac and Regina Piske and Helmut Kettenmann for granting continuous access to their plate reader. We thank the online scientific community for intense discussions, especially Dr. Adam Shapiro (Entasis Therapeutics), and Marius Weismehl for critical reading of the manuscript.

## Author Contributions

**Conceptualization:** Saif Mohd, Edgar Specker, Oliver Daumke.

**Data curation:** Saif Mohd.

**Formal analysis:** Saif Mohd, Edgar Specker, Martin Neuenschwander.

**Funding acquisition:** Saif Mohd, Oliver Daumke.

**Investigation:** Saif Mohd, Andreas Oder, Edgar Specker, Martin Neuenschwander.

**Methodology:** Saif Mohd.

**Project administration:** Jens Peter Von Kries, Oliver Daumke.

**Resources:** Jens Peter Von Kries.

**Supervision:** Oliver Daumke.

**Validation:** Saif Mohd.

**Visualization:** Saif Mohd, Edgar Specker.

**Writing – original draft:** Saif Mohd, Oliver Daumke.

**Writing – review & editing:** Saif Mohd, Edgar Specker, Martin Neuenschwander, Oliver Daumke.

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
