## [Decision Letter · Decision Letter 0]

20 May 2024

PONE-D-24-14329Identification of drug-like molecules targeting the ATPase activity of dynamin-like EHD4PLOS ONE

Dear Dr. Daumke,

Thank you for submitting your manuscript to PLOS ONE. After careful consideration, we feel that it has merit but does not fully meet PLOS ONE’s publication criteria as it currently stands. Therefore, we invite you to submit a revised version of the manuscript that addresses the few points raised during the review process.

We look forward to receiving your revised manuscript.

Kind regards,

Ludger Johannes

Academic Editor

PLOS ONE

“We thank Deutsche Forschungsgemeinschaft (SFB958, project A12, to O.D.) and MDC for their funding and support.”

“We thank Dr. Marc Nazaré, Dr. Peter Lindermann and Jerome Paul for usage of and support

with their Rotavac and Regina Piske and Helmut Kettenmann for granting continuous access

to their plate reader. We thank the online scientific community for intense discussions,

especially Dr. Adam Shapiro (Entasis Therapeutics), and Marius Weismehl for critical reading

of the manuscript. We thank Deutsche Forschungsgemeinschaft (SFB958, project A12, to

O.D.) and MDC for their funding and support.”

“We thank Deutsche Forschungsgemeinschaft (SFB958, project A12, to O.D.) and MDC for their funding and support.”

4. We notice that your supplementary figures and tables are included in the manuscript file. Please remove them and upload them with the file type 'Supporting Information'. Please ensure that each Supporting Information file has a legend listed in the manuscript after the references list.

Reviewers' comments:

Reviewer's Responses to Questions

**Comments to the Author**

1. Is the manuscript technically sound, and do the data support the conclusions?

Reviewer #1: Yes

Reviewer #2: Yes

2. Has the statistical analysis been performed appropriately and rigorously? 

Reviewer #1: Yes

Reviewer #2: Yes

3. Have the authors made all data underlying the findings in their manuscript fully available?

Reviewer #1: Yes

Reviewer #2: Yes

4. Is the manuscript presented in an intelligible fashion and written in standard English?

Reviewer #1: Yes

Reviewer #2: Yes

5. Review Comments to the Author

Reviewer #1: This manuscript describes the development and implementation of a carefully constructed high throughput chemical screen for inhibitors of the EHD family of ATPases. This was not an easy undertaking given the very low intrinsic ATPase activity of this class of enzymes, and the need to assay liposome-stimulated ATPase. The authors have optimized and performed trouble shooting on several steps including:

• Protein expression and purification of the C-terminal ATPase domain of EHD4

• Optimizing of the liposome preparation, including to ensure reproducibility and lack of interference with the malachite-Green reaction

• Defining the minimal concentrations of enzyme and substrate to obtain a workable (albeit suboptimal) signal:noise ratio of 2:1.

In the end they develop an HTP assay with a very respectable Z’ factor of 0.895 and apply this assay to screen ~16,000 compounds from a kinase drug library and the diversity set. They identify 3 compounds and focus on one M8 for preliminary SAR analysis. Despite the rigor, there are a few issues that should be addressed and/or discussed. I believe that these issues can be addressed without further experimentation.

1. Many of the results from enzymatic assays are presented as ‘arbitrary units’ with the background subtracted. Moreover, in all cases the ‘control’ is no enzyme. This makes it difficult to know absolute values and is misleading with respect to signal:noise, which the authors report is a modest 2:1. It would be useful to include at least one example of the raw data.

2. It is unusual for an enzyme activity to be higher at lower temperatures. Do the authors think that this reflects the fluidity of the lipid bilayer? Is this relationship true for the unstimulated activity?

3. The kinetics in Figure 1D are atypical. Why do all of the curves plateau after the first, 5 min, time point? Dynamin’s basal GTPase activity measured using malachite-green is linear for >60 min and >12 min in the presence of liposomes (Leonard et al., Meth. Enzym. 2006). According to the authors the ATP consumption is <10%, so it can’t be substrate limiting, and certainly very little product is produced. Is the enzyme unstable at room temperature? Is this a problem when considering that the screen is conducted for 15 min( i .e. significant inhibition might not alter the amount of Pi produced). The authors should at least comment on this.

BTW, it was difficult to find the incubation time for the screen, I’m assuming it was 15 min based on the legend in Fig. 2D. The incubation time should be clearly indicated in Methods.

4. I was relieved to see that the Michaelis-Menten kinetics were rigorously derived from the experiment in Fig. 2A, where Pi production is linear for 2 min.

5. I’m afraid I’m not familiar with the Bland-Altman method and had to ask ClaudeAI (prompt “how is the Bland Altman method used and interpreted when validating a high throughput enzyme assay). Based on the answer, it’s not clear that yours is the correct use for this method. Moreover, given the variability in positive controls (green dots), I’m a bit surprised that the z’ factor is so high. Certainly the data in Fig 2D can’t be used to calculate z’, but I believe the data in Fig 3B can. Please describe from which data the z’ factor was derived and check to see if you are using the Bland-Altman method correctly.

6. The inhibitors were counter screened for their effects on the malachite green signal in the presence of 30µM Pi (line 264), but under the assay conditions the levels of Pi in the experiment will be much less that 3 µM. Shouldn’t their interference have been tested at lower, more realistic Pi concentrations?

7. Identifying inhibitors for enzymes with very low catalytic activity is difficult, as HTP assays must be performed at very low concentrations of the enzyme and substrate (i.e. at suboptimal conditions). This has hampered the identification of e.g inhibitors of dynamin, as evidenced by the fact that the two best known inhibitors identified in liposome-stimulated screens such as this (dyngo -4a and dynasore) do not inhibit the basal GTPase activity of dynamin (Mohanakrishnan et al., PLoSOne 2017) and indeed their inhibitory effects on endocytosis are independent of dynamin (Park et al. J Cell Sci, 2013). That M8 does not inhibit the liposome-stimulated GTPase activity of DNM1L or EHD2 is a good sign that this compound does not impact liposome integrity; is this true for the other two compounds? Despite the valiant efforts of the investigators, it appears that at the end of the day, the assay yielded only one validated hit, which whose inhibitory effects plateau at 60% inhibition. The authors might indicate in their discussion that other approaches may be needed to identify effective inhibitors of these enzymes.

Reviewer #2: In this manuscript, Mohd et al. describe an in vitro assay of the ATPase activity of EHD4, an important protein involved in cellular processes such as endocytosis and recycling. First, they develop a classical malachite green phosphate sensor assay which measures reliably the ATPase activity of bacterially expressed EHD4(DeltaN), which is stimulated by liposomes containing DOPS. After determining the optimal parameters for ATPase activity measures, they tested ~16000 compounds from the FMP library and found, after screening for false positives, 3 potential specific inhibitors, called MS1, MS8, MS10. Of these, only MS8 was specific for EHD4 (vs EHD2) and did not destabilize the protein, as determined by an absence of shift in melting temperature. Finally, the authors tried to optimize MS8 by substitutions of the main chemical groups but did not find more active compounds. Therefore, MS8 is a first candidate compound to specifically inhibit the ATPase activity of EHD4. Further studies will be required to assess its effect on cellular processes depending on EHD4.

Overall, the study is performed in a well-controlled manner with appropriate data analysis and statistical treatment.

6. PLOS authors have the option to publish the peer review history of their article (what does this mean?). If published, this will include your full peer review and any attached files.

Reviewer #1: **Yes: **Sandra Schmid

Reviewer #2: **Yes: **David PERRAIS

---

## [Author Response · Author response to Decision Letter 0]

30 Jun 2024

We would like to thank Sandy Schmid and David Perrais for the overall positive evaluation and the constructive criticism which has helped to improve our manuscript. Based on the raised issues, we added new data and modified the manuscript at several positions, as indicated in our response letter below. We also corrected a unit mistake in S4D Fig. Changes are marked in blue in the modified manuscript.

Reviewer #1: This manuscript describes the development and implementation of a carefully constructed high throughput chemical screen for inhibitors of the EHD family of ATPases. This was not an easy undertaking given the very low intrinsic ATPase activity of this class of enzymes, and the need to assay liposome-stimulated ATPase. The authors have optimized and performed trouble shooting on several steps including:

• Protein expression and purification of the C-terminal ATPase domain of EHD4

• Optimizing of the liposome preparation, including to ensure reproducibility and lack of interference with the malachite-Green reaction

• Defining the minimal concentrations of enzyme and substrate to obtain a workable (albeit suboptimal) signal:noise ratio of 2:1.

In the end they develop an HTP assay with a very respectable Z’ factor of 0.895 and apply this assay to screen ~16,000 compounds from a kinase drug library and the diversity set. They identify 3 compounds and focus on one M8 for preliminary SAR analysis. Despite the rigor, there are a few issues that should be addressed and/or discussed. I believe that these issues can be addressed without further experimentation.

Thank you very much!

1. Many of the results from enzymatic assays are presented as ‘arbitrary units’ with the background subtracted. Moreover, in all cases the ‘control’ is no enzyme. This makes it difficult to know absolute values and is misleading with respect to signal:noise, which the authors report is a modest 2:1. It would be useful to include at least one example of the raw data.

We now provide the raw data for Fig. 2D without subtracting the background in the modified S2J Fig, including the signal:noise ratio. We chose the 15 min time point for our assay, since the signal:noise was >= 2, while the reaction was still linear (see also Fig. 2D). 

2. It is unusual for an enzyme activity to be higher at lower temperatures. Do the authors think that this reflects the fluidity of the lipid bilayer? Is this relationship true for the unstimulated activity?

As suggested, this observation may relate to different fluidity of the lipid bilayer or to a decreased stability of the EHD4�N construct in the presence of liposomes at higher temperatures. We added a corresponding statement to the modified discussion. 

Compared to the GTPase activity of dynamin, the intrinsic ATPase activity of EHD4ΔN is very low and hardly above the background in this assay, even at high protein concentrations (see new S2H Fig.), which is now also mentioned in the discussion. It was therefore not possible to reliably identify differences in the basal ATPase rates with this assay, induced, for example, by the inhibitors.

3. The kinetics in Figure 1D are atypical. Why do all of the curves plateau after the first, 5 min, time point? Dynamin’s basal GTPase activity measured using malachite-green is linear for >60 min and >12 min in the presence of liposomes (Leonard et al., Meth. Enzym. 2006).

According to the authors the ATP consumption is <10%, so it can’t be substrate limiting, and certainly very little product is produced. Is the enzyme unstable at room temperature? Is this a problem when considering that the screen is conducted for 15 min (i.e. significant inhibition might not alter the amount of Pi produced). The authors should at least comment on this.

Indeed, in the initial conditions, the ATPase activity of EHD4�N plateaued after 5 min, which may relate to a suboptimal protein:liposome concentration used in these experiments. Optimizing this ratio led an almost linear stimulated ATPase rate for the first 15 min (see Fig. 2D). We added these observations and the corresponding dynamin reference to the modified discussion.

BTW, it was difficult to find the incubation time for the screen, I’m assuming it was 15 min based on the legend in Fig. 2D. The incubation time should be clearly indicated in Methods.

The incubation time of 15 min was already mentioned in the methods, but we also added it to the results and figure legend in the modified manuscript (see drug screening section). 

4. I was relieved to see that the Michaelis-Menten kinetics were rigorously derived from the experiment in Fig. 2A, where Pi production is linear for 2 min.

Indeed, for the Michaelis-Menten kinetics, rates were derived from a linear fitting of the initial rates in Fig. 2A. It was mentioned already in the methods and legend of Fig. 2, but we now also added a corresponding statement to the first paragraph of the discussion.

5. I’m afraid I’m not familiar with the Bland-Altman method and had to ask ClaudeAI (prompt “how is the Bland Altman method used and interpreted when validating a high throughput enzyme assay). Based on the answer, it’s not clear that yours is the correct use for this method. Moreover, given the variability in positive controls (green dots), I’m a bit surprised that the z’ factor is so high. Certainly, the data in Fig 2D can’t be used to calculate z’, but I believe the data in Fig 3B can. Please describe from which data the z’ factor was derived and check to see if you are using the Bland-Altman method correctly.

As indicated by ClaudeAI, the Bland-Altman method can be used to assess the agreement between two different measurement methods. However, the original publication (ref. 53 in our manuscript) also indicates that the Bland-Altman method can be used to measure the repeatability of a single measurement method and to derive a repeatability coefficient. Here, we use the Bland-Altman method to obtain such a repeatability coefficient. A condition is that normalized values are used, which is met in our case as we plot the %Activities that were obtained by normalizing to the plate’s positive and negative controls. We acknowledge that some HTS software packages use correlation analysis between two technical replicates as an estimation for repeatability. The original paper also discusses the disadvantages of using correlation analysis for this purpose.

The Z’-factor is calculated on the raw data values of the positive and negative controls of each individual 384-well plate. We use the variant with robust estimators for the control population means and standard deviations (using median for the mean and mean absolute deviation with correction factor for the standard deviations). Then, the mean of Z’-factors of the individual plates is calculated. 

The mean Z’-factor of 0.885 given in Figure 2D was based on the data in Fig. 2E (e.g. 6 plates of the pilot screen). Please note that we corrected a typo (0.895 -> 0.885). The data of the whole screen with 48 plates, including the data of the pilot screen, yield a Z’-factor of 0.827, which is also included in the corresponding figure legend. 

We added some explanation to the Bland-Altman method to the results and provided the calculated Z’ factors to the figure legend with exact reference to the data used for their calculation.

6. The inhibitors were counter screened for their effects on the malachite green signal in the presence of 30 µM Pi (line 264), but under the assay conditions the levels of Pi in the experiment will be much less that 3 µM. Shouldn’t their interference have been tested at lower, more realistic Pi concentrations?

We chose a high Pi concentration for testing the effects of the inhibitors on the colorimetric reaction because of the increased signal:noise compared to lower Pi concentrations. This allowed us to detect even subtle interferences. We added a corresponding statement to the results.

7. Identifying inhibitors for enzymes with very low catalytic activity is difficult, as HTP assays must be performed at very low concentrations of the enzyme and substrate (i.e. at suboptimal conditions). This has hampered the identification of e.g inhibitors of dynamin, as evidenced by the fact that the two best known inhibitors identified in liposome-stimulated screens such as this (dyngo-4a and dynasore) do not inhibit the basal GTPase activity of dynamin (Mohanakrishnan et al., PLoS One 2017) and indeed their inhibitory effects on endocytosis are independent of dynamin (Park et al. J Cell Sci, 2013). That M8 does not inhibit the liposome-stimulated GTPase activity of DNM1L or EHD2 is a good sign that this compound does not impact liposome integrity; is this true for the other two compounds?

Indeed, the two compounds MS1 and MS10 also did not interfere with DNM1L GTPase activity, as we show in the new S4M Fig. We now also explain the reasoning for including such a control in the discussion, including some discussion on possible indirect effects of dynamin inhibitors. 

Despite the valiant efforts of the investigators, it appears that at the end of the day, the assay yielded only one validated hit, which whose inhibitory effects plateau at 60% inhibition. The authors might indicate in their discussion that other approaches may be needed to identify effective inhibitors of these enzymes.

We modified a corresponding statement at the end of the discussion: ‘In future studies, a combined approach of HTS and structure-guided drug design could result in more potent and highly selective EHD4 inhibitor. Our initial SAR studies have indicated preferred chemical modification of MS8 in order to develop such improved inhibitors.’

Reviewer #2: In this manuscript, Mohd et al. describe an in vitro assay of the ATPase activity of EHD4, an important protein involved in cellular processes such as endocytosis and recycling. First, they develop a classical malachite green phosphate sensor assay which measures reliably the ATPase activity of bacterially expressed EHD4(DeltaN), which is stimulated by liposomes containing DOPS. After determining the optimal parameters for ATPase activity measures, they tested ~16000 compounds from the FMP library and found, after screening for false positives, 3 potential specific inhibitors, called MS1, MS8, MS10. Of these, only MS8 was specific for EHD4 (vs EHD2) and did not destabilize the protein, as determined by an absence of shift in melting temperature. Finally, the authors tried to optimize MS8 by substitutions of the main chemical groups but did not find more active compounds. Therefore, MS8 is a first candidate compound to specifically inhibit the ATPase activity of EHD4. Further studies will be required to assess its effect on cellular processes depending on EHD4.

Overall, the study is performed in a well-controlled manner with appropriate data analysis and statistical treatment.

Thank you very much. 

 

Done as requested. 

“We thank Deutsche Forschungsgemeinschaft (SFB958, project A12, to O.D.) and MDC for their funding and support.”

If this statement is not correct you must amend it as needed. Please include this amended Role of Funder statement in your cover letter; we will change the online submission form on your behalf.

"The funders had no role in study design, data collection and analysis, decision to publish, or preparation of the manuscript." We added a corresponding statement to the cover letter.

“We thank Dr. Marc Nazaré, Dr. Peter Lindermann and Jerome Paul for usage of and support with their Rotavac and Regina Piske and Helmut Kettenmann for granting continuous access to their plate reader. We thank the online scientific community for intense discussions, especially Dr. Adam Shapiro (Entasis Therapeutics), and Marius Weismehl for critical reading of the manuscript. We thank Deutsche Forschungsgemeinschaft (SFB958, project A12, to O.D.) and MDC for their funding and support.” We note that you have provided funding information that is currently declared in your Funding Statement. However, funding information should not appear in the Acknowledgments section or other areas of your manuscript. We will only publish funding information present in the Funding Statement section of the online submission form.

“We thank Deutsche Forschungsgemeinschaft (SFB958, project A12, to O.D.) and MDC for their funding and support.”

We removed the funding statement from the Acknowledgement and added a corresponding statement to the cover letter.

4. We notice that your supplementary figures and tables are included in the manuscript file. Please remove them and upload them with the file type 'Supporting Information'. Please ensure that each Supporting Information file has a legend listed in the manuscript after the references list.

Done as requested.

We carefully re-checked the reference list.

---

## [Decision Letter · Decision Letter 1]

15 Jul 2024

Identification of drug-like molecules targeting the ATPase activity of dynamin-like EHD4

PONE-D-24-14329R1

Dear Dr. Daumke,

We’re pleased to inform you that your manuscript has been judged scientifically suitable for publication and will be formally accepted for publication once it meets all outstanding technical requirements.

Kind regards,

Ludger Johannes

Academic Editor

PLOS ONE

Additional Editor Comments (optional):

Reviewers' comments:

Reviewer's Responses to Questions

**Comments to the Author**

1. If the authors have adequately addressed your comments raised in a previous round of review and you feel that this manuscript is now acceptable for publication, you may indicate that here to bypass the “Comments to the Author” section, enter your conflict of interest statement in the “Confidential to Editor” section, and submit your "Accept" recommendation.

Reviewer #1: All comments have been addressed

2. Is the manuscript technically sound, and do the data support the conclusions?

Reviewer #1: Yes

3. Has the statistical analysis been performed appropriately and rigorously? 

Reviewer #1: Yes

4. Have the authors made all data underlying the findings in their manuscript fully available?

Reviewer #1: Yes

5. Is the manuscript presented in an intelligible fashion and written in standard English?

Reviewer #1: Yes

6. Review Comments to the Author

Reviewer #1: (No Response)

7. PLOS authors have the option to publish the peer review history of their article (what does this mean?). If published, this will include your full peer review and any attached files.

Reviewer #1: No

---

## [Editor Report · Acceptance letter]

19 Jul 2024

PONE-D-24-14329R1 

PLOS ONE

Dear Dr. Daumke, 

I'm pleased to inform you that your manuscript has been deemed suitable for publication in PLOS ONE. Congratulations! Your manuscript is now being handed over to our production team.

Kind regards, 

on behalf of

Dr. Ludger Johannes 

Academic Editor

PLOS ONE